# CBL-Interacting Protein Kinases 18 (*CIPK18*) Gene Positively Regulates Drought Resistance in Potato

**DOI:** 10.3390/ijms24043613

**Published:** 2023-02-10

**Authors:** Liang Yang, Ning Zhang, Kaitong Wang, Zhiyong Zheng, Jingjing Wei, Huaijun Si

**Affiliations:** 1College of Life Science and Technology, Gansu Agricultural University, Lanzhou 730070, China; 2State Key Laboratory of Aridland Crop Science, Gansu Agricultural University, Lanzhou 730070, China; 3College of Agronomy, Gansu Agricultural University, Lanzhou 730070, China

**Keywords:** potato, StCIPK18, drought stress, protein interaction

## Abstract

Sensor—responder complexes comprising calcineurin B-like (CBL) proteins and CBL-interacting protein kinases (CIPKs) are plant-specific Ca^2+^ receptors, and the CBL-CIPK module is widely involved in plant growth and development and a large number of abiotic stress response signaling pathways. In this study, the potato cv. “Atlantic” was subjected to a water deficiency treatment and the expression of *StCIPK18* gene was detected by qRT-PCR. The subcellular localization of StCIPK18 protein was observed by a confocal laser scanning microscope. The StCIPK18 interacting protein was identified and verified by yeast two-hybrid (Y2H) and bimolecular fluorescence complementation (BiFC). *StCIPK18* overexpression and *StCIPK18* knockout plants were constructed. The phenotypic changes under drought stress were indicated by water loss rate, relative water content, MDA and proline contents, and CAT, SOD and POD activities. The results showed that *StCIPK18* expression was upregulated under drought stress. StCIPK18 is localized in the cell membrane and cytoplasm. Y2H shows the interaction between StCIPK18 and StCBL1, StCBL4, StCBL6 and StCBL8. BiFC further verifies the reliability of the interaction between StCIPK18 and StCBL4. Under drought stress, *StCIPK18* overexpression decreased the water loss rate and MDA, and increased RWC, proline contents and CAT, SOD and POD activities; however, *StCIPK18* knockout showed opposite results, compared with the wild type, in response to drought stress. The results can provide information for the molecular mechanism of the StCIPK18 regulating potato response to drought stress.

## 1. Introduction

Potato (*Solanum tuberosum*) is one of the four major food crops in the world. It is widely grown in many countries and regions around the world. With global climate change increasingly aggravated, the frequency and duration of abiotic stresses, such as drought, salt damage and extreme temperature, are increasing day by day, leading to lower yields and impaired quality in the world [1]. As an important second messenger, Ca^2+^ plays a significant role in the signal transduction of plant hormones and various stress responses in plant cells. Up to now, many Ca^2+^ receptor proteins have been reported, mainly including Calmodulins (CaMs) family, Calmodulin-like proteins (CMLs) family, Ca^2+^ dependent protein kinases (CDPKs) and Calcineurin B-like proteins (CBLs). Among them, the CBL, as a unique Ca^2+^ receptor in the plants, plays an important role in response to Ca^2+^ signals. CBL protein without kinase activity cannot function alone. It needs to be combined with Calcineurin B-like interacting protein kinase (CIPK) to form a complex to complete Ca^2+^ signal decoding and transmission [2]. CIPK is a plant-specific serine-threonine protein kinase, and its protein structure contains an N-terminal kinase domain and a C-terminal regulatory domain. The CIPK interacts with the CBL proteins through the highly conserved domain of C-terminal asparagine-alanine-phenylpropanamide (NAF), and they form a complex that acts on specific target proteins, thus participating in the stress response of plants to various abiotic stresses [3].

Relevant studies have shown that CIPK plays an important regulatory role in plant growth, development and abiotic stress tolerance, among which the most classical one is the CBL-CIPK complex mediated salt overly sensitive (SOS) signal transduction pathway [4]. The SOS signaling pathway was first identified in *Arabidopsis thaliana* in 1996, and the core members of this pathway are AtCBL4 (SOS3), AtCIPK24 (SOS2) and Na^+^/H^+^ antiporter (SOS1) [5]. AtCBL4(SOS3) -AtCIPK24 (SOS2) complex phosphorylates activate Na^+^/H^+^ antiporter at the plasma membrane, removing excess Na^+^ from cells, thereby improving plant salt tolerance [6]. Under low K^+^ stress, AtCBL1/AtCBL9 localized AtCIPK23 to the plasma membrane and subsequently phosphorylated AtCIPK23 to activate AKT1. Activated AKT1 enhanced the absorption of K^+^ by cells at low K^+^ concentration [7,8]. In addition, the expression of *AtCIPK6* is induced by salt and drought stress, and overexpression of *AtCIPK6* in *Arabidopsis* can enhance the tolerance of *Arabidopsis* to salt stress, while increasing the sensitivity to abscisic acid (ABA) [9]. *AtCIPK24* mutant plants can reduce transpiration and water loss rate in plant leaves and show stronger drought tolerance. AtCBL1/9 can interact with AtCIPK24 and target AtCIPK24 to plasma membrane. Moreover, AtCBL1/9-AtCIPK24 complex simultaneously regulates K^+^ transport in root and stomatal guard cells [10]. In rice (*Oryza sativa*), OsCBL1-OsCIPK23 was identified as an upstream regulator of OsAKT1, which enhances OSAKT1-mediated K^+^ uptake by rice roots [11]. The heterologous expression of maize (*Zea mays*) ZmCIPK3 in *Arabidopsis* enhanced its drought tolerance [12]. Cold stress can induce upregulation expression of *OsCIPK3*, and overexpression of *OsCIPK3* can improve the tolerance of rice to cold stress [13]. TaCIPK23 can interact with TaCBL1 on the cytoplasmic membrane, and overexpression of *TaCIPK23* can improve drought resistance and the germination rate of wheat [14]. In potatoes, overexpression of the *StCIPK10* gene can enhance the ability of scavenging reactive oxygen species and the tolerance of potato to drought and osmotic stress [15].

The *CIPK* gene family in the model plant *Arabidopsis* has been systematically identified, and there have been many reports on the function of individual *CIPK* genes. However, the expression patterns and specific functions of *CIPK* genes in potatoes under different stress and hormone treatments remain to be elucidated. We have systematically identified the *CIPK* gene family members in potatoes in the previous work [15]. In this study, the potato *StCIPK18* gene was cloned, the tissue-specific expression of *StCIPK18* was verified by real-time quantitative polymerase chain reaction (qRT-PCR) and the expression site of StCIPK18 in cells was determined by subcellular localization. The resistance of transgenic lines to drought stress was verified by using *StCIPK18* overexpression lines and interfered expression lines. We used yeast two-hybrid (Y2H) to screen the CBL protein interacting with StCIPK18 and to verify its reliability by bimolecular fluorescence complementation (BiFC). The results of this study provide reference for further research on improving potato quality and resistance.

## 2. Results

### 2.1. Bioinformatics and Expression Analysis of StCIPK18

The bioinformatics analysis results of the potato *StCIPK18* gene showed that the length of gene coding region was 1413 bp, encoding 470 amino acids. According to NCBI Splign analysis, the *StCIPK18* gene did not contain introns, indicating that it was a continuous gene. The gene is located on chromosome 9 (PGSC gene ID: Soltu.DM.09G010080.1; and Chromosome location: chr09: 13195990–13198651). The StCIPK18 amino acid sequence was entered into the SMART online website and the analysis of its conserved domain showed that StCIPK18 contains the typical serine/threonine kinase domain and the NAF motif. Physical and chemical properties of the protein showed that the molecular weight of the protein was 53.03624KD, PI 8.59, indicating that the protein was a basic protein. The hydrophilic coefficient was −0.394, indicating that the protein was hydrophilic. MEGA7.0 software was used to analyze the evolutionary relationship between StCIPK18 and other plants. The results showed that StCIPK18 was closely related to *Solanum commersonii* (Appendix A), which was consistent with the law of biological evolution. To initially understand the possible regulatory mechanism of the *StCIPK18* gene, PlantCARE was used to analyze the 2000 kb sequence upstream of the gene promoter (Table 1). The results showed that the gene had 13 basic gene expression control elements, followed by six light-responsive elements and two hormone-responsive elements, ABRE and TCA-element. Finally, there were three abiotic stress response elements, among which MBS is the MYB binding site involved in drought induction, indicating that this gene may potentially participate in drought stress response.

According to the results of qRT-PCR (Figure 1A), the expression level of the *StCIPK18* gene in potato tissues had significant differences (*p* < 0.05). The relative expression level of this gene was lowest in tubers, but the relative expression level was higher in stems and leaves; they were 55 times and 45 times higher in tubers, respectively. To analyze the expression level of *StCIPK18* under drought treatment, potato leaves after drought treatment were collected for expression analysis (Figure 1B). The results showed that *StCIPK18* expression was significantly upregulated when the soil water content was lower than that of the control group, and the highest *StCIPK18* expression was 6.19 times more than the control group when the soil water content was 35~40%.

### 2.2. Cloning and Subcellular Localization of StCIPK18 in Potatoes

The leaf cDNA of potato cv. “Atlantic” was used as the template for PCR amplification, and 1% agarose gel electrophoresis was used for detection. The results showed that the size of the *StCIPK18* gene fragment was about 1413 bp (Appendix A), which was as expected.

The pCAMBIA1300-35S-EGFP empty vector was used as a negative control, and pCAMBIA1300-EGFP-StCIPK18 was transiently expressed in tobacco leaf cells and observed under a laser confocal scanning microscope. The results showed that in the control group, the GFP signal was expressed in the cell membrane, cytoplasm and nucleus of tobacco cells, without specific division. However, in the experimental group, pCAMBIA1300-EGFP-StCIPK18 fusion protein was mainly expressed in the cell membrane and cytoplasm, but not in the nucleus (Figure 2).

### 2.3. Interaction between StCIPK18 and StCBLs

CBL and CIPK family members can interact with each other to form complexes, which play different roles in plant response to different abiotic stresses, and their combinations show diversity and complementarity. Ten potato *StCBLs* genes (*StCBL1*, *StCBL3*, *StCBL4*, *StCBL6*, *StCBL7*, *StCBL8*, *StCBL10*, *StCBL11*, *StCBL12* and *StCBL13*) were cloned and connected to pGADT7 vector. The BD-StCIPK18 bait vector and pGBKT7 were transformed into AH109 yeast cells, respectively. Meanwhile, the control group was also transferred into yeast cells and coated on defective medium, respectively, for protein interaction analysis. The yeast of each combination could grow normally on the medium SD-Leu-Trp, indicating that the recombinant plasmid was successfully transferred into yeast cells and had no toxicity to yeast strains. The strains co-transformed with the recombinant plasmids could not grow on SD-Trp-Leu-His-Ade, indicating no self-activation (Figure 3A). On the defective medium SD-Trp-Leu-His-Ade, the combinations StCIPK18-StCBL1, StCIPK18-StCBL4, StCIPK18-StCBL6 and StCIPK18-StCBL8 showed colony growth, while the negative control showed no colony growth (Figure 3B). Moreover, further verification on the medium supplemented with X-α-gal showed that the colonies of the positive control group and the experimental group turned blue, indicating that the reporter gene β-galactosidase gene z (β-Galactosidase gene z, LacZ) was activated and expressed functional β-galactosidase. The results showed that StCIPK18 could interact with StCBL1, StCBL4, StCBL6 and StCBL8, respectively. To further verify the results of Y2H, StCBL4 was selected for the next step of the experiment. The recombinant plasmids pSPYCE-StCIPK18 and pSPYNE-StCBL4 were injected into tobacco leaf cells by *Agrobacterium tumefaciens* for the transient expression experiment. The signal of yellow fluorescent protein was observed in tobacco cells under confocal laser microscopy, which demonstrated the interaction between StCIPK18 and StCBL4 (Figure 3C).

### 2.4. Genetic Transformation of StCIPK18 Gene in Potatoes and Identification of Transgenic Plants

The potato cv. “Atlantic” microtubers were infected in vitro with *Agrobacterium* solution containing pCAMBIA1300-35S-StCIPK18 overexpression vector and pCPB121-StCIPK18 down-expression vector, respectively, and cultured on a differentiation medium to form callus and differentiated buds (Figure 4A–D). Rooted transgenic plants were screened by a rooting medium containing hygromycin and kanamycin, respectively. The transformed plants that successfully developed roots were identified by amplification of the reporter gene *HPT*, on the overexpression vector pCAMBIA1300-35S-StCIPK18, and the reporter gene *NPTII*, on the down-expression vector pCPB121-StCIPK18, respectively. Genomic DNA from the transformed plants was extracted by CTAB and analyzed using PCR and agarose gel electrophoresis. The plants transformed with pCAMBIA1300-35S-*StCIPK18* overexpression vector successfully amplified 598 bp *HPT* gene, and the plants downregulated with pCPB121-StCIPK18 successfully amplified 676 bp *NPTII* gene (Figure 4E,F). However, in the wild plants, neither of the two reporter genes could be successfully amplified, which was in line with our experimental expectations.

Total RNA was extracted from non-transgenic and transgenic plants and reverse transcribed into cDNA. The relative expression level of *StCIPK18* in both plants was analyzed by qRT-PCR (Figure 4G,H). The results showed that the relative expression level of *StCIPK18* in overexpressed plants OE-1, OE-2 and OE-3 was significantly higher than that in non-transgenic plants, which were 17.7 times, 18.8 times and 8.1 times higher, respectively. However, in the downregulated expression plants RNAi-1, RNAi-2 and RNAi-3, the relative expression levels of RNAi-1, RNAi-2 and RNAi-3 were significantly lower than those of non-transgenic plants, which were 0.29 times, 0.33 times and 0.28 times higher than those of non-transgenic plants, respectively. Therefore, these results prove that overexpression and down-expression transgenic plants have been successfully obtained.

### 2.5. StCIPK18 Positively Regulates Potato Drought Stress Resistance

To explore the function of *StCIPK18* in drought tolerance, WT, OE-n and RNAi-n lines were subjected to water deficiency treatment for 14 days. There was no difference in the growth status of each line before water deficiency treatment; however, after treatment, the growth status of the OE-n line was significantly better than that of the WT and RNAi-n lines, and the leaves of RNAi-n lines were obviously wilting (Figure 5A). Water loss rate and RWC can reflect drought resistance, to a certain extent. After drought treatment, OE-n lines had a higher water content and lower water loss rate than WT lines, but the results of RNAi-n lines were just the opposite (Figure 5B,C). The activities of three key antioxidant enzymes CAT, SOD and POD were determined. The results showed that the enzyme activities of CAT, SOD and POD were not significantly different in WT, OE-n and RNAi-n lines before treatment; however, the activities of these three enzymes increased greatly in OE-n lines, more than in WT lines after drought treatment, and had opposite results in RNAi-n lines (Figure 5D–F). There was no significant difference in Pro and MDA content between the lines before treatment. After treatment, the Pro content in OE-n lines was significantly higher than that in WT, and the MDA content was lower. The results in RNAi-n lines were opposite (Figure 5G,H).

## 3. Discussion

The potato is an important food crop worldwide, and drought stress can seriously affect its yield and quality [16]. In this study, we focused on the role of *StCIPK18* in potato response to drought stress and analyzed its regulatory mechanism. We found that this gene has cis-acting elements in response to drought, and drought stress can increase the transcription level of *StCIPK18*. Therefore, *StCIPK18* related strategies can be considered to improve the stress resistance of potato plants. Furthermore, we identified and validated upstream StCBLs proteins that interact with StCIPK18.

In the Ca^2+^ signal transduction pathway, the CBL-CIPK complex, as a plant-specific Ca^2+^ receiver and decoder, plays an important role in maintaining the normal growth and development of plants and in regulating the stress response. In order to verify whether *StCIPK18* is differentially expressed under drought conditions, potato plants were subjected to drought treatment, and the expression level of *StCIPK18* was subsequently analyzed by qRT-PCR. Drought conditions led to a significant upregulation of *StCIPK18.* Subcellular localization experiments can determine the location of proteins within the cell and provide clues to the potential interaction between proteins and other proteins. Moreover, CBL can also pull CIPK, which interacts with it, to where it is required to function. In this study, StCIPK18 was localized in the cytoplasm and cell membrane by observing the green fluorescence signal. Therefore, it is speculated that StCIPK18 plays a role in cell signal transduction. Studies have shown that AtCIPK1 and AtCIPK24 in *Arabidopsis thaliana* are also expressed in cell membrane and cytoplasm [17,18]. As for the influence of StCIPK18 protein localization on its specific function and signaling pathway, it is necessary to make *StCIPK18* gene stable expression in the plant and identify its interacting proteins, and then conduct further study.

CBL-CIPK module is a large and complex system, and different combinations of the two members may participate in different signaling pathways and play different roles. In this study, the interaction between potato StCIPK18 and 10 StCBLs was verified by yeast two-hybrid. The results show that StCIPK18 interacts with StCBL1, StCBL4, StCBL6 and StCBL8. This suggests that StCIPK18 may play a role in several different signaling pathways. Subsequently, we verified the reliability of the interaction between StCIPK18 and StCBL4 through BiFC. The results showed that yellow fluorescence signal was found on the tobacco cell membrane, indicating the interaction between StCIPK18 and StCBL4. The reliability of the specific interaction of other combinations still needs to be verified further. CIPK binds to different CBLs to function at different sites; for example, AtCIPK24/SOS2 can interact with AtCBL4/SOS3 and AtCBL10, respectively, but their biological functions are performed in the plasma membrane and vacuole membrane, respectively [6,19]. Furthermore, AtCBL4 was specifically expressed in the root of the plant, while AtCBL10 played a role in the aboveground part [20]. AtCBL1 and AtCBL9 can interact specifically with AtCIPK1 under drought conditions and then respond in an ABA-dependent and ABA-independent manner, respectively, to regulate cellular osmotic balance [21,22]. These results not only indicate that CBLs can target CIPKs to different sites to perform specific biological functions, but also indicate that the interaction and mechanism between CBLs and CIPKs are complex and diverse. In order to understand the specific stress response pathway of *StCIPK18* in potatoes, we need to analyze and verify the role and function of StCIPK18 and its interaction with CBL proteins.

To understand the role of *StCIPK18*, potato plants with an overexpression of *StCIPK18* or disturbed expression of *StCIPK18* were developed and subjected to water deficiency treatment. RWC is an important parameter for plants to cope with drought stress, and good water status can represent the tolerance of plants to drought, osmotic and high salt stress [23,24]. After water deficiency treatment, the RWC of OE-StCIPK18 was significantly higher than that of WT and RNAi; on the contrary, OE-StCIPK18 had lower water loss rate than RNAi-StCIPK18. Drought stress can increase the level of ROS and the damage of cellular structures, and thus, reduce the tolerance of plants to stress [25]. In response to drought, plants also use antioxidant enzymes, including CAT, SOD and POD [26]. The experimental results showed that CAT, SOD and POD activities were increased under drought stress, and the antioxidant enzyme activity of the OE-StCIPK18 strain was significantly higher than that of WT, while the results of the RNAi-StCIPK18 strain were opposite, indicating that the antioxidant enzyme activity was increased in OE-StCIPK18 strains. This keeps them from being poisoned by ROS under drought stress, thus improving the drought resistance of the potato plant. MDA content is usually used to show oxidative damage, and proline, as an osmotic regulator, plays a role in maintaining osmotic pressure and stabilizing cell membrane structure. Therefore, the two are widely regarded as oxidation markers and osmotic markers, respectively [27]. We found that the MDA content of OE-StCIPK18 lines were significantly lower than that of WT under drought stress, while that of RNAi-StCIPK18 lines were higher than that of WT. On the contrary, drought could induce the accumulation of Pro, and the accumulation of OE-StCIPK18 lines were significantly higher than that of WT, while that of RNAi-StCIPK18 lines were the opposite. This further proves that overexpression of *StCIPK18* in potatoes can improve potato drought tolerance. Consistent results have also been found in other plants that overexpression of *OsCIPK10* in rice can improve its drought resistance [28]. Heterologous expression of *TaCIPK2* in tobacco can improve the drought resistance of plants by improving the ability to remove ROS and regulating the expression of stress-related genes [29]. These two genes were clustered as homologous genes of *StCIPK18* in the previous study of Ma [15]. Specific environmental signal pressures can induce specific interactions between CBL and CIPK, in response to stress, by activating phosphorylation of downstream proteins. For example, the AtCBL1/9-AtCIPK23-AKT1 pathway in *Arabidopsis* indirectly improves drought resistance by closing stomata in plant leaves through K^+^ absorption, which enhances the water-holding capacity of the leaves [30]. TaCBL3-TaCIPK29 enhances salt tolerance in wheat by regulating the transporter gene and antioxidant system of wheat [31]. These results suggest that the *CIPK* gene can regulate plant stress tolerance in a variety of ways.

In summary, our results indicate that *StCIPK18* can play an important role in potato drought stress response and defense by improving ROS clearance ability. Combined with the experimental results, a pathway model of *StCIPK18* response to drought stress was proposed (Figure 6). Drought could regulate the expression of *StCIPK18* and improve the ability of potatoes to remove ROS. Drought can also affect the intracellular Ca^2+^ concentration, specifically guiding StCBL proteins to form a dimer with StCIPK18, thus regulating the expression of downstream drought response genes, and directly or indirectly improving the drought resistance of plants. However, whether StCBL1, StCBL4, StCBL6 and StCBL8 can specifically bind StCIPK18 and further regulate plant drought resistance under drought stress remains unclear, and further experiments are needed to prove it. Therefore, quantitative analysis and functional identification of StCIPK18 interacting CBL proteins under drought stress are necessary to study the signaling pathway and functional mechanism of StCIPK18 in response to drought stress.

## 4. Materials and Methods

### 4.1. Growth Conditions and Treatment of Plant Materials

For the acquisition of microtubers, potato cv. “Atlantic” seedlings were cut into stem segments with one leaf, inoculated in a solid MS medium containing 8% sucrose and cultured for 28 days at 22 °C and a 16 h light/8 h dark cycle in a light incubator. Then, they were transferred to dark conditions for culture for 45–60 days to obtain the microtubers [32].

“Atlantic” seedlings were seeded in an MS medium containing 3% sucrose and incubated in a light incubator under 22 °C, with a 16 h light/8 h dark cycle. After 21 days, the seedlings were transplanted into 10 × 10 cm plastic pots containing a 3:1 (V: V) mixture of nutritive soil and vermiculite, kept at 24 °C with a 16 h light/8 h dark cycle and 60% relative humidity conditions to develop [33] 28 days in the greenhouse of Gansu Agricultural University. For tissue specificity analysis, the roots, stems and leaves of “Atlantic” were collected. For tubers, the microtubers cultured under dark conditions were harvested as samples. For tuber buds, the microtubers were placed at (23 ± 2) °C in the dark for sprouting. When the tuber buds grew to 0.2 cm, a sample with a diameter of 0.8 cm was used for sampling. The above operations were repeated three biological times. These samples were quickly frozen in liquid nitrogen and stored at −80 °C for standby. We selected the plants with the same height and health status among the remaining plants and applied drought treatment; soil water content in the pots was kept at 75–80% in the control group (CK), 55–60% in water stress group 1 (WS1), 35–40% in WS2 and 15–20% in WS3. Soil water content was monitored at 10:00 and 16:00 every day using TDR-300 sensors (Spectrum R, Aurora, Illinois, USA). The third to fifth leaves from the top of the potato plants were collected for quantification of *StCIPK18*, and three biological replicates were performed [34]. After freezing in liquid nitrogen, the leaves were stored at −80 °C for later use.

Tobacco (*Nicotiana Benthamian*) was cultured in a pot culture of 10 × 10 cm with nutrient soil vermiculite (V: V = 1:1). The growth condition was 2000 Lx light intensity, a 16 h light/8 h dark cycle and the temperature was (23 ± 2) °C. Samples were cultured for four to five weeks for relevant experiments.

### 4.2. Bioinformatics Analysis

The *StCIPK18* (ID: Soltu.DM.09G010080.1) sequence was retrieved in potato database Spud DB. The physicochemical properties of the StCIPK18 protein were analyzed by ProtParamtool. The divergence of exon–intron structures was analyzed using NCBI Splign. The domains of the protein were analyzed using the online tools at the SMART website. DNAMAN6 was used for protein sequence analysis. Based on the neighbor joining method, MEGA7.0 software was used to construct the phylogenetic tree of potato StCIPK18. PlantCARE was used to analyze the cis-acting elements in the upstream 2000 bp of the promoter region of *StCIPK18*.

### 4.3. Cloning of StCIPK18

For total RNA extraction and first-strand cDNA synthesis, please refer to the manual of TRNzol Universal Total RNA extraction Kit and FastKing gDNA Dispelling RT SuperMix reverse transcription kit (Tiangen, Beijing, China), respectively. The CDS region of the *StCIPK18* gene was cloned using the leaf cDNA of the potato variety “Atlantic” as a template. The reaction system was as follows: OE-StCIPK18-F 1 μL, OE-StCIPK18-R 1 μL, cDNA template 1 μL, PrimerSTAR HS (Premix) (Takara, Dalian, China) 10 μL, ddH2O 7 μL. The reaction conditions were as follows: 95 °C for 5 min, followed by 34 cycles of 95 °C for 30 s, 58.7 °C for 30 s, 72 °C for 90 s and finally, 72 °C for 10 min. The primer sequences are shown in Appendix A.

### 4.4. StCIPK18 Expression Analysis by qRT-PCR

Total RNA extraction and first-strand cDNA synthesis used the same methods as previously described. SYBR^®^ Green Premix Pro Taq HS qPCR KIT (ROX Plus) (Accurate Biology, Changsha, China) was used to analyze the expression level of *StCIPK18*. Polymerase chain reaction (PCR) solution (20 μL) contained 10 μL 2 × 2 × SYBR^®^ Green Pro Taq HS Premix, 0.4 μL forward and reverse primers, 1 μL cDNA (100 ng) template and 8.2 μL nuclease-free water. qRT-PCR was performed in the Light Cycler 96 system (Roche, Diagnostics GmbH, Basel, Switzerland) with the following parameters: 95 °C for 30 s, followed by 40 cycles of 95 °C for 5 s and 60 °C for 30 s, and the *StEF1α* (GenBank ID: AB061263.1) gene was used as a standardized reference gene. The primer sequences are shown in Appendix A. All experiments were performed with three biological replicates and three technical replicates. The relative expression levels of genes were calculated by the 2^−ΔΔCt^ method [35].

### 4.5. Subcellular Localization of StCIPK18

Subcellular localization vectors were constructed by the homologous recombination method. Specific primers were designed by the coding region of the *StCIPK18* gene and vector sequence of pCAMBIA1300-35S-EGFP. The encoding region of the *StCIPK18* gene, without stop codon, was amplified using the cDNA template of potato cv. “Atlantic”. The PCR product was inserted into the vector pCAMBIA1300-35S-EGFP with restriction sites *Kpn* I and *Xba* I; the recombinant vector pCAMBIA1300-EGFP-StCIPK18 and empty vector pCAMBIA1300-35S-EGFP were then transferred into *Agrobacterium* GV3101(Angyubio, Shanghai, China). Tobacco at 4 to 5 weeks of age were selected for subcellular localization experiment. A small hole was inserted into the back of the 2nd to 4th tobacco leaves with a sterile syringe and the *Agrobacterium* solution was slowly injected into the tobacco leaves. The excess bacterial solution was wiped off with filter paper and marked. The tobacco infected with *Agrobacterium* was cultured in the dark at 25 °C for 1 day and then transferred to light for 1 to 2 days. EGFP fluorescence signals were observed using a laser scanning confocal microscope (CARI ZEISS, LSCM 800, Oberkochen, Baden-Württemberg, Germany) at a laser wavelength of 488 nm.

### 4.6. Yeast Two-Hybrid Assay

Specific primers were designed according to the sequences of the *StCIPK18* gene and pGBKT7 vector, as well as the sequences of *StCBL* genes and the pGADT7 vector. The primer sequences were shown in Appendix A, and the *StCBL* gene information was referred to Cai et al. [36]. The target fragment was amplified by PCR. Homologous recombination method was used to connect the coding region of the *StCIPK18* gene to the pGBKT7 vector, which contained the restriction sites of *Nde* I and *Sal* I, and the coding region of *StCBL* genes to the pGADT7 vector, which contained the restriction sites of *Nde* I and *Sac* I. The BD-StCIPK18 recombinant vector and AD-StCBLs recombinant vector were obtained. For the self-activation experiment, the BD-StCIPK18 vector and empty plasmid pGADT7, AD-StCBLs vectors and empty plasmid pGBKT7 were co-transformed into yeast AH109 (Angyubio, Shanghai, China), respectively, to observe their growth status. Using pGADT7-Rect+ pGBKT7-53 as a positive control and pGADT7-Rect+ pGBKT7-Lam as a negative control, the BD-StCIPK18 vector and AD-StCBLs vectors were transformed into yeast AH109 receptor cells by LiAc/PEG4000 mediated method. The bacterial solution was coated on the SD-Leu-Trp medium, incubated at 30 °C for 2 days and the positive colonies were selected and diluted in 10 μL 0.9% NaCl solution. For 4 days, 5 μL bacterial solution samples were absorbed and cultured on SD-Leu-Trp-His medium at 30 °C. After that, positive colonies were selected again and diluted in 10 μL 0.9% NaCl solution. For 4 days, 5 μL bacterial solution was absorbed and sampled on the culture medium of SD-Leu-Trp-His-Ade and SD-Leu-Trp-His-Ade + X-α-gal, respectively, grew at 30 °C and was photographed. All experiments were performed according to the protocol described in the Yeast Transformation System User’s Manual (Clontech).

### 4.7. Bimolecular Fluorescence Complementarity Assay

BiFC was used to verify the reliability of the Y2H results. Specific primers were used to amplify the coding region sequences of *StCIPK18* and *StCBL4* genes without stop codon. PCR products were connected to pSPYCE-35S and pSPYNE-35S with restriction sites of *Xba* I and *Kpn* I, respectively, and the recombinant plasmids pSPYCE-StCIPK18 and pSPYNE-StCBL4 were obtained. The recombinant plasmid was transformed into *Agrobacterium* GV3101(Angyubio, Shanghai, China); after re-suspension, the bacterial solutions containing pSPYCE-StCIPK18 were mixed with the bacterial solutions containing pSPYNE-StCBL4 at a ratio of 1:1. pSPYCE-StCIPK18+pSPYNE-35S was used as a negative control, and the mixed bacterial solution of each combination was injected into tobacco leaves using sterile syringes. After 48–72 h, the yellow fluorescent protein signal (YFP) was observed using laser confocal scanning electron microscopy (CARI ZEISS, LSCM800, Zeiss, Oberkochen, Baden-Württemberg, Germany).

### 4.8. Construction of Plant Expression Vectors

Specific primers were designed based on *StCIPK18* CDS sequence and vector pCAMBIA1300-35S-EGFP sequence. The target product was amplified by PCR using specific primers and linked to the vector pCAMBIA1300-35S-EGFP, which contained restriction sites *Kpn* I and *Xba* I by homologous recombination. After PCR amplification, double restriction enzyme digestion and sequencing, the overexpression vector containing the correct sequence was named pCAMBIA1300-35S-StCIPK18. For downregulated expression vectors, Oligo from the online site WMD3 was used to design precursor primers (I, II, III and IV) (Appendix A) and obtain the target fragment by standard PCR [37]. The PCR product was ligated into the pMD™ 18-T vector (TaKaRa Bio, Beijing, China). After PCR detection, double digestion and sequencing verification, *Sac* I and *Kpn* I were used to double enzyme digest the correct recombinant vector, and T4 ligase was used to link the target fragment to the linearized pCPB121 vector containing *Sac* I and *Kpn* I restriction sites (Figure 7). The recombinant plasmid verified by PCR and sequencing was named pCPB121-StCIPK18. The constructed recombinant plasmid was transferred to *Agrobacterium* GV3101(Angyubio, Shanghai, China) [38].

### 4.9. Genetic Transformation of Potatoes and Identification of Transgenic Plants

Microtubers of potato cv. “Atlantic” were used for genetic transformation experiments. We used a sterile blade to cut the microtubers into slices with a thickness of 0.3–0.5 cm. Then, we put the microtuber slices into a triangle bottle with 50 mL liquid *Agrobacterium*, containing recombinant plasmid, to infect for 7–10 min. During this period, the triangle bottle was shaken constantly, and then we wiped the residual bacteria liquid on the microtuber slices with sterile filter paper. Then, the microtuber slices were co-cultured on a solid MS medium containing 3% sucrose at 28 °C in the dark for 2 days. After 2 days, the microtuber slices were further cultured on a new solid MS medium at 25 °C, 2500 Lx ray, and the medium was changed once a week [39]. When the differentiated shoots reached 1–1.5cm, they were cut and moved to the rooting medium containing kanamycin (75 mg/L) or hygromycin (50 mg/L) and carbenmycin (200 mg/L), respectively, for rooting screening [40] using the CTAB method to extract the DNA of transgenic plants that could root normally. Non-transferred plants were used as a negative control and empty vector pCAMBIA1300-35S-EGFP was used as a positive control. Specific primers HPT-F and HPT-R, NPTII-F and NPTII-R were used to amplify the 598 bp gene fragment of hygromycin B phosphotransferase (*HPT*) and 676 bp gene fragment of neomycin phosphotransferase (*NPT II*) gene fragment, respectively. Plants whose target fragments are amplified are named OE-n and RNAi-n, respectively. In addition, qRT-PCR was used to detect the expression of the *StCIPK18* gene in transgenic potato plants for further identification for subsequent experiments.

### 4.10. Drought Stress Treatment

Wild type potato plants and transgenic plants were inoculated in an MS solid medium with 3% sucrose content and cultured in an incubator for 21 days. Then, the plants were transferred to pots and continued to grow for 4 weeks (16 h light/8 h dark, 24 °C). Then, we selected the plants with the same growth state and healthy state for subsequent treatment. For the water loss rate experiment, leaves of 4-week-old plants were selected and placed in plastic petri dishes and exposed to the air in the greenhouse for 5 h. The loss of fresh weight of leaves was measured every hour. Three plants were selected from each line and three biological replicates were made for each plant. For the drought treatment, plants were treated with water deficiency; after 14 days, photos were taken and samples were collected. For the RWC experiment, the third to fourth whole leaves of plants were collected before and after drought treatment and the fresh weight of the collected leaves was measured immediately. The leaves were placed in distilled water at room temperature overnight, then the excess water on the leaves was absorbed with paper and the saturated weight was measured. The expanded leaves were dried in the oven at 105 °C for 6–8 h and the dry weight of the leaves was measured [41]. RWC (%) = [(fresh weight) − (dry weight)/(saturated weight) − (dry weight)] × 100%. MDA content was determined using the thiobarbiturate method described by Heath and Packer [42], Pro content was determined using the method described by Bate et al. [43]. SOD activity was determined using the method previously described by Giannopolitis and Ries [44] and POD activity was determined using the method previously described by Maehly and Chance [45]. The CAT activity can be measured in H2O2 decomposition; specific methods reference Aebi [46]. In the above experiments, 3 biological replicates and 3 technical replicates were performed for each strain.

### 4.11. Statistical Analyses

Microsoft Excel 365 was used to sort out the data and SPSS Statistics 22 (IBM, Chicago, Illinois, USA) was used for statistical analysis. Kolmogorov Smirnov normality test and Levene’s homogeneity of variance test were used for the evaluation before the statistical analysis, and all data conformed to normal distribution and showed homoscedasticity. Results were analyzed using data variance analysis performed with the ANOVA Duncan’s test. Significance was defined as significant (*) at *p* < 0.05 and highly significant (**) at *p* < 0.01.

## 5. Conclusions

In this study, the potato *StCIPK18* gene was cloned. qRT-PCR analysis showed that the *StCIPK18* gene was expressed in different tissues, with the highest relative expression level in stem and leaves, and the *StCIPK18* gene was induced by drought stress. Subcellular localization showed that StCIPK18 was localized in the cell membrane and cytoplasm. The interaction between StCIPK18 and StCBL1, StCBL4, StCBL6 and StCBL8 is verified by Y2H, and the reliability of the interaction between StCIPK18 and StCBL4 is tested by BiFC. Overexpression of *StCIPK18* could reduce the water loss rate and MDA content, increase the activities of antioxidant enzymes and proline content and thus improve the tolerance of plants to drought stress. These results can provide theoretical basis for the molecular mechanism of potato response to drought stress.

## Figures and Tables

**Figure 1 ijms-24-03613-f001:**
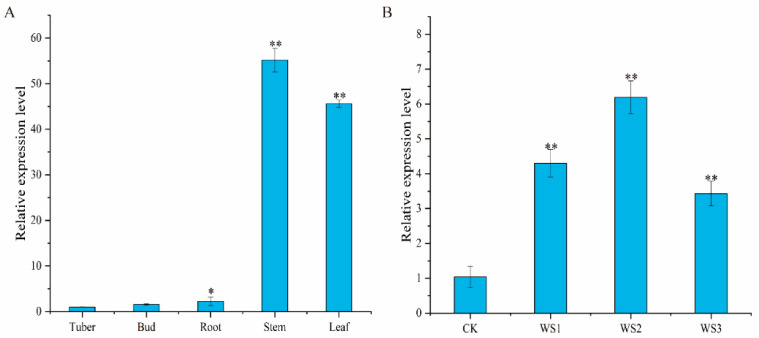
Analysis of potato *StCIPK18* gene expression. (**A**) The relative expression level of *StCIPK18* in different organs of potatoes. (**B**) Relative expression of the *StCIPK18* gene under drought stress. CK: control; WS1: water stress group 1; WS2: water stress group 2; WS3: water stress group 3. Relative expression levels, determined by qRT-PCR, relative to the expression of the *StEFla* gene, are expressed as 2^−ΔΔCt^. Each column represents the mean values ± SE (*n* = 3; * *p* < 0.05; ** *p* < 0.01).

**Figure 2 ijms-24-03613-f002:**
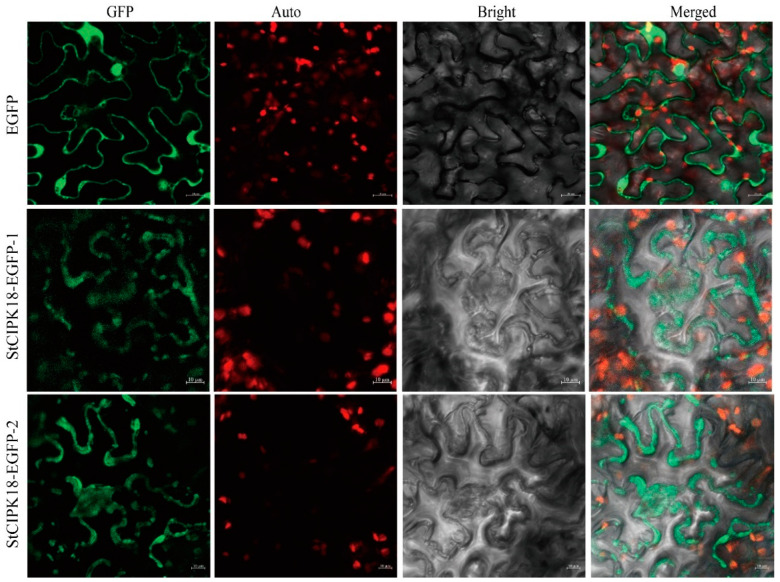
Subcellular localization of the StCIPK18 protein in tobacco leaf cells. The EGFP and StCIPK18-EGFP fusion protein transiently expressed in tobacco. GFP: EGFP fluorescence signal in the dark field; Auto: Autofluorescence of chlorophyll; Bright: Cell morphology under bright field; Merged: Combination field. The scale bale represents 10 μm.

**Figure 3 ijms-24-03613-f003:**
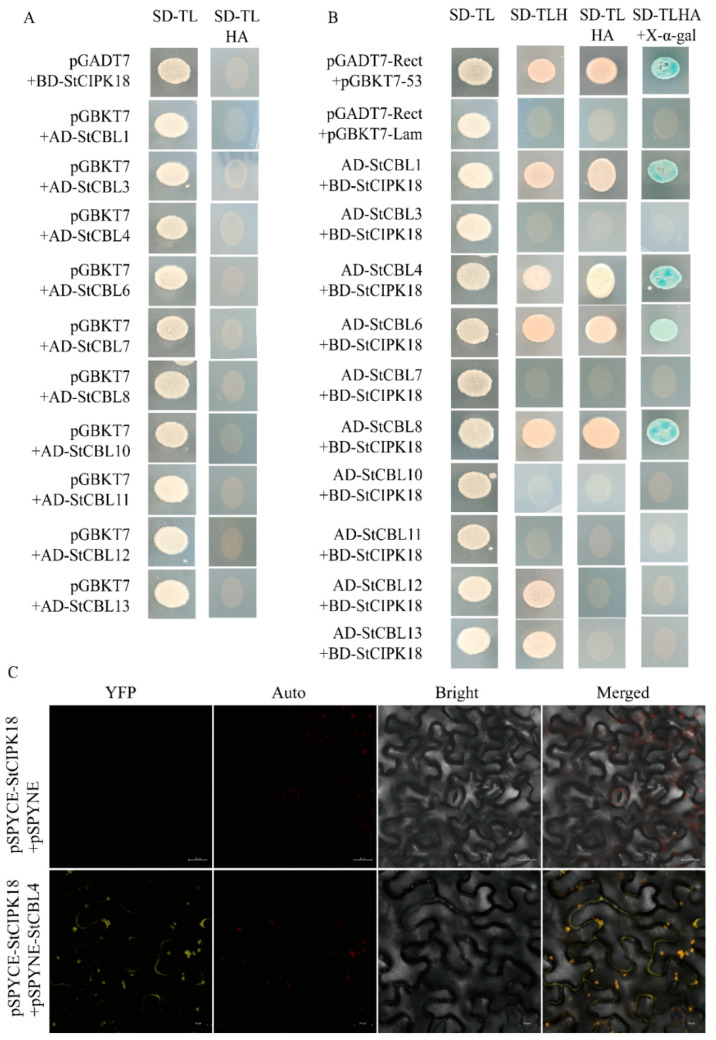
Interaction between StCIPK18 and StCBLs, demonstrated by yeast two-hybrid assay and BiFC. (**A**) Self-activation experiment of StCIPK18 and StCBLs. (**B**) Y2H assay analyzing the interaction between StCIPK18 (bait) and StCBLs (prey). (**C**) BiFC assay of the interaction between StCIPK18 and StCBL4. StCIPK18 was introduced into the pSPYCE vector and fused with C-terminal YFP; StCBL4 was introduced into the pSPYNE vector and fused with N-terminal YFP. pSPYCE-StCIPK18 + pSPYNE was used as a negative control. The scale bale represents 20 μm.

**Figure 4 ijms-24-03613-f004:**
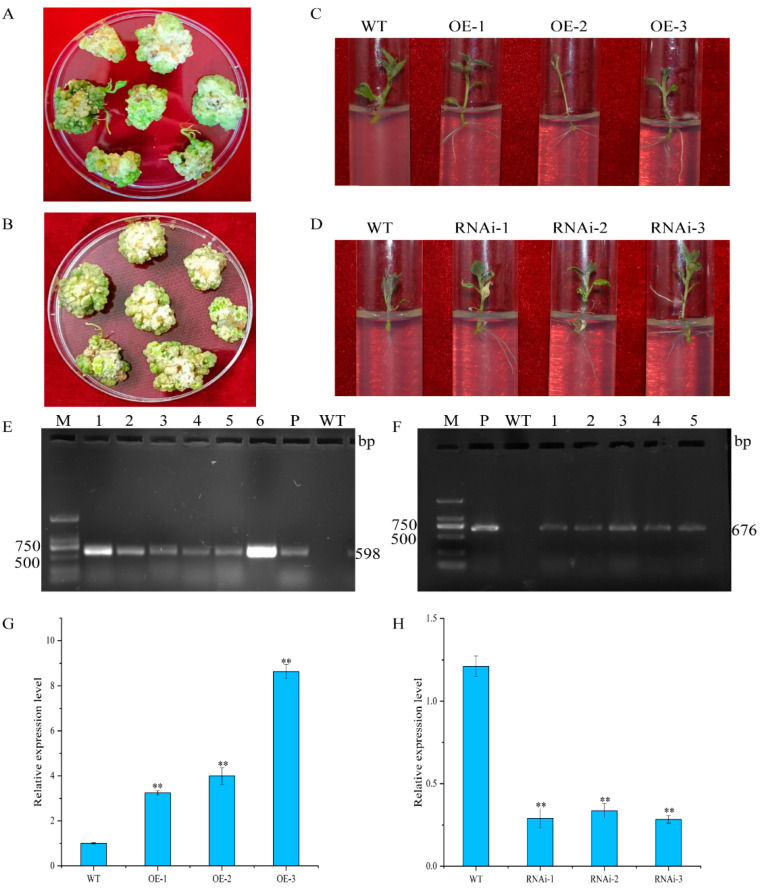
Acquisition and identification of transgenic plants. (**A**,**B**) Callus and differentiated buds. (**C**,**D**) Rooting and screening transgenic plants. WT: Wild-type plant “Atlantic”; OE: Transgenic plant “Atlantic” carrying recombinant plasmids pCAMBIA1300-35S-StCIPK18; RNAi: Transgenic plant “Atlantic” carrying recombinant plasmids pCPB121-StCIPK18. (**E**,**F**) PCR detection of transgenic plants. M: DL 2000 marker; P: Positive control plasmid; WT: Negative control; 1–6: Transgenic lines. (**G**,**H**) The relative expression level *StCIPK18* in the transgenic plants and WT plants. WT: Wild-type plants of “Atlantic”; OE-1~OE-3: Transgenic plants of “Atlantic” carrying recombinant plasmids pCAMBIA1300-35S-StCIPK18; RNAi-1~RNAi-3: Transgenic tubers of “Atlantic” carrying recombinant plasmids pCPB121-StCIPK18. Each column represents the mean values ± SE (*n* = 3; ** *p* < 0.01).

**Figure 5 ijms-24-03613-f005:**
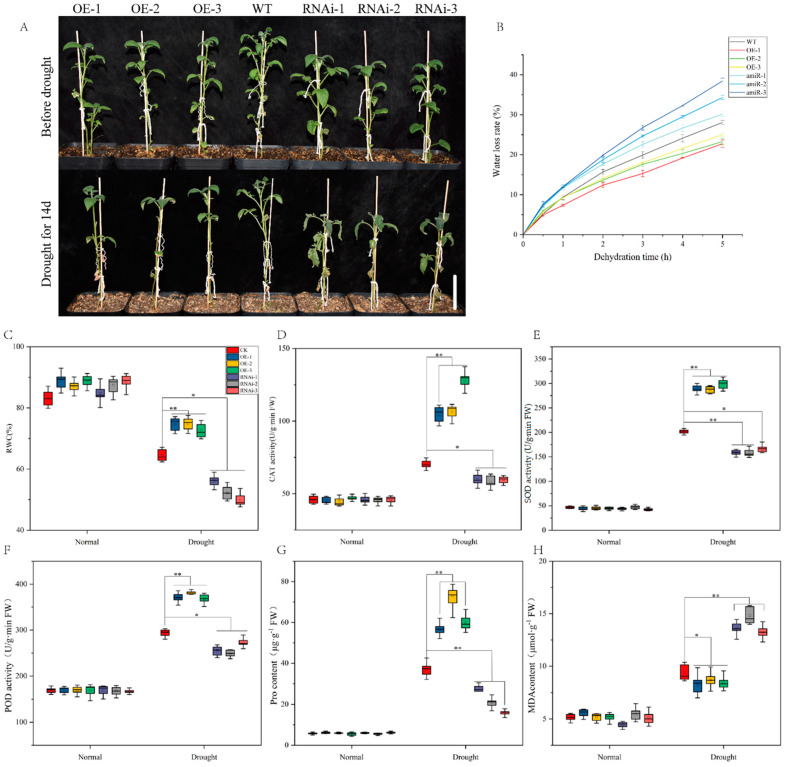
Overexpression of *StCIPK18* improved drought tolerance after 14 days of water deficiency. (**A**) Phenotypic differences between WT, OE-StCIPK18 and RNAi-StCIPK18 lines after 14 days of water deficiency. The scale bale represents 10 cm. (**B**) Water loss from detached leaves. (**C**) Leaf relative water content under non-stress and drought conditions. (**D**) CAT activity. (**E**) SOD activity. (**F**) POD activity. (**G**) Proline content. (**H**) MDA content. Each column represents the mean values SE (*n* = 3; * *p* < 0.05; ** *p* < 0.01).

**Figure 6 ijms-24-03613-f006:**
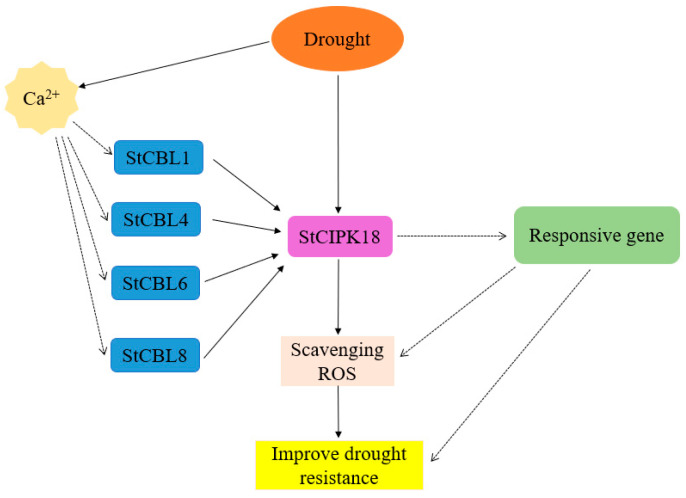
Prediction pathway model of potato *StCIPK18* response to drought stress. *StCIPK18* regulates the ability of plants to remove ROS to improve drought tolerance. CBLs can receive Ca^2+^ signal and then combine with StCIPK18 to form a complex, which can regulate the drought resistance of plants by regulating the expression of downstream stress-responsive genes and improving the activity of antioxidant enzymes. The solid line indicates the pathways that have been tested to have a direct effect and the dashed line indicates the pathways that have an indirect effect and require further verification.

**Figure 7 ijms-24-03613-f007:**
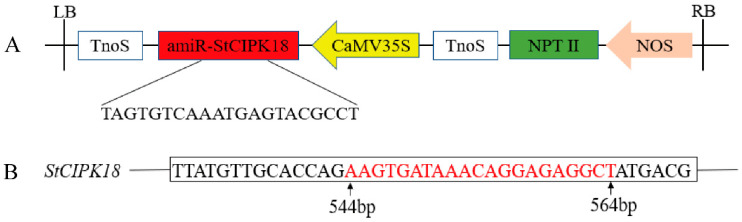
RNAi-mediated gene silence in transgenic potato plants. (**A**) Schematic illustration of the engineered pCPB121-StCIPK18 vector. (**B**) Schematic illustration of the target region of the *StCIPK18* gene.

**Table 1 ijms-24-03613-t001:** *Cis*-acting element analysis.

Name	Sequence	Number	Function
ABRE	ACGTG	1	*cis*-acting element involved in the abscisic acid responsiveness
ARE	AAACCA	2	*cis*-acting regulatory element essential for the anaerobic induction
CAAT-box	CAAAT	13	common *cis*-acting element in promoter and enhancer regions
MBS	CAACTG	1	MYB binding site involved in drought-inducibility
TC-rich repeats	GTTTTCTTAC	1	*cis*-acting element involved in defense and stress responsiveness
TCA-element	CCATCTTTTT	1	*cis*-acting element involved in salicylic acid responsiveness
TCCC-motif	TCTCCCT	1	part of a light responsive element
TCT-motif	TCTTAC	1	part of a light responsive element
ATC-motif	AGCTATCCA	1	part of a conserved DNA module involved in light responsiveness
Box 4	ATTAAT	2	part of a conserved DNA module involved in light responsiveness
G-Box	CACGTT	1	*cis*-acting regulatory element involved in light responsiveness
Sp1	GGGCGG	1	light responsive element

## Data Availability

The datasets used and/or analyzed during the current study are available from the corresponding author on reasonable request.

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
