# Peer review of "CBL-Interacting Protein Kinases 18 (CIPK18) Gene Positively Regulates Drought Resistance in Potato"

_ijms, 2023, doi:10.3390/ijms24043613_

Round 1

Reviewer 1 Report

The manuscript entitled “CBL-Interacting protein Kinase 18 gene positively regulates drought resistance in potato” by Yang et al aimed to study the expression of the gene StCIPK18 and the interacting protein by using a confocal laser scanning microscope. 

The authors found that the expression of gene StCIPK18 was up-regulated under drought stress, and the protein localizes in the cell membrane and cytoplasm, interacting with StCBL4. The overexpression of StCIPK18 showed decreased water loss rate and MDA, increased RWC and proline contents and CAT, SOD and POD activities. The study is very interesting, and provides direct evidence to support potato resistance response. 

Overall, the method used in the study is thorough. Conclusions are appropriate, and supported by the data. Statistical analysis is provided within the manuscript. The whole study is sound, and I recommend accepting it.

Author Response

Thank you very much for reviewing our manuscript and give positive suggestions. We revised the manuscript method and Figure 5.

Reviewer 2 Report

Dear editor and colleagues,

I have read with interest the manuscript ‘CBL-Interacting Protein Kinases 18 (CIPK18) Gene Positively Regulates Drought Resistance in Potato’ submitted to IJMS-mdpi.

It is a study that describes the interaction of CIPK-CBL proteins of potato and elucidates their role for abiotic stress tolerance. The paper has affinity to the aims and scopes of IJMS, is rather well designed and executed, uses appropriate techniques and the results support the conclusions. Hence, I believe the work has merit for publication

Specifically:

Introduction

             the background provided is relevant to the work

             the literature cited is up to date

             it includes a clear hypothesis and aims

             Is well-structured and logical in progression

Materials and Methods

             the study has been appropriately designed

             adequate controls have been used

             the authors have appropriately justified their choices

             the methods adequately described except the statistics

Results, Data and Figures

             data presentation is appropriate and comprehensible

             data are interpreted accurately

             titles and legends of figures/tables are accurately descriptive

Discussion and Conclusion

             are well-structured and logical in progression

             the conclusions are supported by the data presented

Shortcomings

·         I urge the authors to include a paragraph regarding anova, post hoc analyses test and test for normality (that is a prerequisite for anova)

·         In figure3, Y2H pictures are very bright; enhancement is needed

·         My personal opinion is that bars only tell half of the story. I prefer boxplots since they provide a preview of data distribution. Since the authors used 3 biological replicates and 3 technical replicates (9 datapoints) I think boxplots should provide a better picture.

Based on the above I recommend a minor revision

Author Response

Reponse:

1. I urge the authors to include a paragraph regarding anova, post hoc analyses test and test for normality (that is a prerequisite for anova).

First of all, thank you for pointing out the problem.  We have added this part as required, and the specific content is " Microsoft Excel 365 was used to sort out the data and SPSS Statistics 22(IBM, Chicago, IL, USA) was used for statistical analysis. Kolmogorov Smirnov normality test and Levene's homogeneity of variance test were used for evaluation before statistical analysis, and all data conformed to normal distribution and showed homoskedness. Results were analyzed using data variance analysis performed with the ANOVA Duncan’s test. Significance was defined as significant (*) at p < 0.05 and highly significant (**) at p < 0.01".

2. In figure3, Y2H pictures are very bright; enhancement is needed.

In the Y2H picture, although both positive and negative plaques were bright, there were still obvious differences. I think it may be due to the amount of bacteria inhaled during sampling. In fact, the negative colonies did not grow properly on the the culture medium of SD-Leu-Trp-His-Ade and SD-Leu-Trp-His-Ade + X-α-gal until they dried up, while the positive colonies did not. Therefore, no modification was made.

3. My personal opinion is that bars only tell half of the story. I prefer boxplots since they provide a preview of data distribution. Since the authors used 3 biological replicates and 3 technical replicates (9 datapoints) I think boxplots should provide a better picture.

Figure 5 has been remaken following suggestions..